# Preparation and Characterization of Developed Cu_x_Sn_1−x_O_2_ Nanocomposite and Its Promising Methane Gas Sensing Properties

**DOI:** 10.3390/s19102257

**Published:** 2019-05-16

**Authors:** Nagih M. Shaalan, Dalia Hamad, Abdullah Aljaafari, Atta Y. Abdel-Latief, Mostafa A. Abdel-Rahim

**Affiliations:** 1Physics Department, Faculty of Science, King Faisal University, P.O. Box 400, Al-Hassa 31982, Saudi Arabia; 2Physics Department, Faculty of Science, Assiut University, Assiut 71516, Egypt; noniatypes@yahoo.com (D.H.); atta552001@yahoo.com (A.Y.A.-L.); maabdelrahim@yahoo.com (M.A.A.-R.)

**Keywords:** Cu_x_Sn_1−x_, CuO/SnO_2_, heterojunction, optical properties, methane, sensing mechanism

## Abstract

Novel materials with nanostructures are effective in controlling the physical properties needed for specific applications. The use of active and sensing materials is increasing in many applications, such as gas sensing. In the present work, we attempted to synthesize incorporated Cu^2+^ into the SnO_2_ matrix as Cu_x_Sn_1−x_O_2_ nanocomposite using a cost-effective precursor and method. It was observed that, at low concentrations of copper precursor, only SnO_2_ phase could be detected by X-ray diffraction (XRD)_._ The distribution of Cu in the SnO_2_ matrix was further measured by elemental analysis of energy-dispersive X-ray (EDX) mapping and X-ray fluorescence (XRF). At high copper concentration, a separated monoclinic phase of CuO was formed (noted here as CuO/SnO_2_). The average crystallite size was slightly reduced from 5.9 nm to 4.7 nm with low doping of 0.00–5.00% Cu but increased up to 15.0 nm at high doping of 10.00% Cu upon the formation of separated SnO_2_ and CuO phases. The formation of Cu–SnO_2_ or CuO phases at low and high concentrations was also observed by photoluminescent spectra. Here, only the emission peak of SnO_2_ with a slight blueshift was recorded at low concentrations, while only the CuO emission peak was recorded at high concentration. The effect of Cu concentration on the sensing properties of SnO_2_ toward methane (CH_4_) gas was also investigated. It was found that the sensor embedded with 2.00% Cu exhibited an excellent sensitivity of 69.0 at 350 °C and a short response–recovery time compared with the other sensors reported here. The sensing mechanism of Cu_x_Sn_1−x_O_2_ and CuO/SnO_2_ is thus proposed based on Cu incorporation.

## 1. Introduction

Improvement of binary oxide nanomaterials to detect gas with high sensitivity and precise limitation requires creative materials. To date, SnO_2_ has attracted the attention of researchers for gas sensing applications. It is one of the most significant n-type nanomaterials with a wide band gap [1]. Its sensing property is enhanced in the form of nanoscale due to the increased surface area and defect structure, which have higher reactivity [2].

To improve the sensitivity of sensing materials, nanoadditives, such as Pd, Pt, Au, Ag, and Cu, have been employed [3]. SnO_2_ in pure or doped phase with Ag and Fe elements exhibits a low sensitivity toward CH_4_ [4,5,6,7]. Several studies have focused on the effect of doping materials or nanoadditives of noble metals to improve the sensing properties of SnO_2_ toward CH_4_. The sensing properties of SnO_2_ toward CH_4_ are improved by up to 1.55/0.1% (response/gas concentration) and 4.5/0.1% with Pt doping, as reported in [8,9]. Wagner et al. [10] and Cabot et al. [11] reported an improvement in the sensing properties of SnO_2_ upon the functionalization with Pd, with the sensitivity increasing up to 20/0.66% at 400 °C [10]. Fateminia et al. [12] reported a response toward methane with Pt–SnO_2_ structure as high as 30/0.5%. Much improvement was also reported when binary, ternary, or quadrant composite-based SnO_2_ were developed [13,14,15,16].

The doping of SnO_2_ nanostructures by Cu^2+^ has been shown to enhance the gas sensing properties of SnO_2_. The role of Cu doping in SnO_2_ sensing properties toward H_2_S was studied by Wei et al. [17]. Zhang et al. [18] found that sensors fabricated of Cu-doped porous thin film showed a good performance toward H_2_S. Ultrasensitive room-temperature H_2_S gas sensing prepared by Cu^2+^-doped SnO_2_ was also presented in [19]. Cu-doped SnO_2_ was recently studied for an ethanol gas sensor. The sensing behavior offered a suitable application of Cu-doped SnO_2_ nanowires with 2.5 at.% Cu in the nanowires [20]. Other promising single-phase materials, such as CuO, Co_3_O_4_, and In_2_O_3_, have also been shown to exhibit a sensing performance toward CH_4_, although they are still in need of improvement [21,22,23]. A comparison between several materials sensitive to methane is reported in Table 1. 

The response of reducing gases, such as CH_4_, is dependent on the space charge region of the n-type oxide, where it reduces as the level of gas increases. Obviously, an increase in the depth of the depletion layer will significantly enhance the response of the oxide toward the reducing gas. Yamazoe [24] reported that the trivalent impurities of Al^3+^ in SnO_2_ generated more donor concentrations beneath the oxide surface, resulting in an increase in the depletion thickness from ~3 nm to ~19 nm. However, the pentavalent doping of Sb^5+^ caused an increase in the electron density, resulting in a decrease in the thickness of the depletion layer. 

In this paper, we attempted to achieve high sensitivity of the SnO_2_ nanostructure toward CH_4_ using metallic doping with cost-effective materials and a cheap preparation method. To this end, we synthesized a developed nanomaterial based on the incorporation of Cu^2+^ in the matrix of SnO_2_. The developed nanostructures were investigated by X-ray diffraction (XRD), X-ray fluorescence (XRF), energy-dispersive X-ray (EDX), and high-resolution transmission electron microscopy (HRTEM). Spectrofluorometric analysis was used to emphasize the formation of Cu_x_Sn_1−x_O_2_ and separated CuO phase. The gas sensing performance toward CH_4_ was systematically investigated for a sensing layer fabricated of this nanomaterial. A gas-sensing mechanism is thus proposed for CuO/SnO_2_ and Cu_x_Sn_1−x_O_2_. 

## 2. Experimental Details

For the synthesis of Cu_x_Sn_1−x_O_2_ nanostructures, tin(IV) chloride [1*−x*: SnCl_4_·5H_2_O] or copper chloride dehydrate [*x*: CuCl_2_·2H_2_O] as a dopant source were dissolved in 10.0 mL of distilled water, followed by a drop-wise addition of 3.0 g NaOH, which had previously been dissolved in 10 mL water. The reactants were prepared based on the molar ratio of [CuCl_2_·2H_2_O (*x*) + SnCl_4_·5H_2_O (1−*x*) + 4NaOH] for several concentrations of *x*: 0.00%, 0.25%, 0.50%, 1.00%, 2.00%, 5.00%, and 10.00%. The mixture was stirred at room temperature and then transferred into Teflon-lined stainless steel autoclaves maintained at 160 °C for 10 h. Figure 1 shows the change in product color caused by increasing concentration of CuCl_2_·2H_2_O in order to evaluate the effect of Cu on the SnO_2_ powder. A dramatic change in color was observed at *x* = 10.00%.

The obtained powder was scanned by XRD (Philips PW 1700, Cu-Kα) in the range of 15° to 90° with a speed of 0.06°/s. The crystallinity and morphology were examined by TEM (JEOL-JEM-2100F). Fluorescence (FL) spectra were recorded at room temperature using a spectrofluorometer (Jasco, FP-6300 WRE) with an excitation wavelength of 290 nm and 600 nm. 

For sensor fabrication, two gold electrodes were deposited using DC sputtering on a glass substrate. It was then coated by the sensing materials using a screen-printing method with a thickness of about 20 µm. The gas sensor was thermally treated at 400 °C for 30 min in the ambient air before inserting it into the testing chamber. The operating temperature started at room temperatures of up to 400 °C, controlled by programmable temperature controller Omega-CN4300 with autotuning function. Dry synthetic air mixed with CH_4_ gas was passed through the measurement chamber at a rate of 200 mL/min, controlled by Horiba mass flow controllers (SEC-N112MGM). The electrical measurements were carried out using a computerized data acquisition instrument (multichannel Agilent 34972A LXI). The sensor response was defined by the ratio of electrical resistance, R_g_/R_a_, where Ra is the resistance in synthetic air, and R_g_ is the resistance in CH_4_.

## 3. Results and Discussions

### 3.1. Effect of Concentration Ratio on Crystal Structure and Morphology

To emphasize the incorporation of Cu in SnO_2_, XRF and EDX techniques were used. The elemental analysis of the nanoparticles by XRF spectroscopy and the actual mole percentage of Cu and Sn are listed in Table 2. The molar fractions, determined from the XRF results, were slightly less than those of the intended values. This behavior may be attributed to the difference in the solubility constant of the precursors. Figure 2a illustrates XRD charts of the prepared powder. For pure SnO_2_, the diffraction peaks were in agreement with the tetragonal structure of SnO_2_ reported in JCPDS no. 77-0452. As the concentration of *x* increased from 0.25% to 5.00%, no peaks were observed for either Cu or CuO phase, but a shift was observed in peaks with lower diffraction angle. This indicated that Cu ions may have been embedded into the crystal lattice of SnO_2_, thereby causing lattice expansion. A secondary phase of monoclinic CuO was detected at 10.00%, matching with the reported values in JCPDS card no. 04-015-5876. To investigate the effect of *x* concentration on the crystal structure, peaks of (110) and (101) are shown in the expanded scale in Figure 2b. The position of (110) and (101) peaks slightly shifted to a lower value of the diffraction angle for Cu_x_Sn_1−x_O_2_. This proved that Cu caused an expansion of the SnO_2_ lattice. On the other hand, when the *x* concentration was increased to 10.00%, SnO_2_ and CuO peaks were observed at the exact position. Although the *x* concentration was only 10.00%, the black color of CuO dominated the color of the composite. The average crystallite size was calculated using the Scherrer equation [25]:(1)$$D=\frac{0.9\;\lambda }{\beta \cos\theta }$$
where λ is the X-ray wavelength, β is the full width at half maximum, and θ is the corresponding angle. The crystallite size was 5.9–4.6 nm for the concentration range of *x* = 0.25–5.00%, but it was 15.0 nm for x = 10.00%. The crystallite size and lattice parameters are listed in Table 3. The lattice parameters were calculated using Celref UNITCELL refinement software with the raw data measured by XRD. The slight decrease in *D* may be attributed to the incorporation of Cu in the host matrix, which enabled more nucleation sites and slowed down the growth of the crystals [26]. By comparing the ionic radii of Sn^4+^ and Cu^2+^, substitution was found to be the most dominant mechanism in the incorporation of Cu^2+^ ions into the SnO_2_ lattice at low concentration, with the ionic radii of O^2−^, Sn^4+^, and Cu^2+^ being 1.40, 0.69, and 0.73 Å, respectively. This result was supported by the slight expansion in unit cells as well as by the EDX data for selected samples of 0.50%, 2.00%, and 10.00% Cu, as shown in Table 2 and Table 3. EDX mapping exhibited uniform distribution for O, Sn, and Cu, as shown in Figure 3. EDX of OK, SnL, and CuK showed that the weight percentage of Cu increased with increasing Cu precursor. In addition, the ratio of Cu atomic to Sn was observed as 0.092, 0.283, and 0.282, corresponding to 1.368, 0.575, and 1.219 oxygen atom ratio to Cu and Sn atoms. We interpreted this as follows: (1) at low incorporation of 0.50% Cu, the bridging oxygen was assigned partially to Cu and largely to Sn ions (SnO_2_), and the oxygen ratio was thus 1.368; (2) when the Cu was increased up to 2.00%, the bridging oxygen was missing because Cu was involved in the lattice, and the atomic oxygen ratio therefore decreased to 0.575; (3) upon the formation of CuO as a separated phase, more oxygen ions were linked with both Cu^2+^ and Sn^4+^, thus forming CuO and SnO_2_, respectively. Murray et al. reported the effect of stoichiometry and the absence of bridging oxygen on the structure of TiO_2_ [27].

The structural information when *x* = 2.00% and *x* = 10.00% was further studied using HRTEM. The morphology of the 2.00% structure was mostly spherical, while it was nanorod-like for the 10.00% structure when the CuO phase was formed, as shown in Figure 4a–f. A spherical-like structure with an average crystallite or grain size of 4.52 nm was observed for *x* = 2.00%. The interplanar spacing of 0.330 nm matched with (110) of the tetragonal SnO_2_ phase. For *x* = 10.00%, the data revealed the presence of both SnO_2_ and CuO phases with the spacing of 0.334 and 0.252 nm, corresponding with (110) for SnO_2_ and (002) plane for CuO, and also an increase in the crystallite size. 

### 3.2. Effect of Concentration Ratio on Fluorescence Spectra 

The FL technique has been widely used to investigate the structures and defects of metal oxides. FL spectrum was recorded at λ_ex_ = 290 and λ_ex_ = 600 nm excitation wavelengths in order to detect the radiative recombination in SnO_2_ and CuO, respectively, as shown in Figure 5. Nonradiative recombination is defined by a cross-relaxation process where two neighboring ions exchange energy [28]. The undoped SnO_2_ showed peaks at 335 and 383 nm. These were attributed to the radiative recombination of the free exciton (recombination of conduction electrons of Sn5p to the hole of O2p) and to electron transition into defect levels, such as oxygen vacancies and the luminescence centers. A very slight blueshift and intensity reduction were observed for the emission peak of SnO_2_. At the excitation energy of λ_ex_ = 600 nm, only the emission peak of CuO appeared at 806 nm for high concentration of 10.00% Cu, as shown in the inset of Figure 5. The energy of 1.54 eV refers to the phase of CuO. It is ascribed to the radiative recombination of Cu conduction band to its valance band. These two peaks confirmed the formation of separated phases of CuO/SnO_2_.

### 3.3. Sensing Mechanism of the Composites

An increase in the depletion layer of oxide is very useful for the reducing reaction, which can be achieved by increasing the vacancies or surface area. Thus, the importance of vacancies and the surface area appears when the ambient oxygen molecules chemically adsorb on the surface and capture free electrons from the conduction band of the oxide [29,30]. These oxygen molecules form O^−^, O_2_^−^, and O^2−^ species based on the surface temperature, which results in a decrease/increase of free carriers inside n-type/p-type oxide [31,32]. For reducing gases such as CH_4_, the gas molecule reacts with the adsorbed oxygen species and produces H_2_O and CO_2_. Upon this reaction, the electrons return to the oxide conduction band, leading to a change in its conductivity. In low concentrations of *x*, the incorporation of Cu^2+^ into the SnO_2_ matrix caused lattice tension because of the larger ionic radii of Cu^2+^, as detected by the lattice parameter expansion. It raised the donor density in SnO_2_, allowing the adsorbed oxygen molecules to pick up the excess electrons, which increased the depletion layer. The average crystallite size of Cu_x_Sn_1−x_O_2_ slightly decreased, while the excessive Cu caused a formation of a separated CuO phase. Nanorod-like structure with large crystallite size was grown in the presence of CuO phase, resulting in a decrease in the surface area as well as the thickness of the depletion layer compared to the bulk thickness. The formation of CuO resulted in CuO/SnO_2_ heterojunction, but this heterojunction was insignificant compared to the effect of the bulk size. The expected band structure between SnO_2_ and CuO is shown in Figure 6a. The exact band structure of CuO/SnO_2_ was plotted using AFORS-Het software for the given parameters in the figure. Based on this band structure and the result obtained from the below gas sensing measurements, we can report the following. When the reducing gas CH_4_ was introduced into a sensing layer composited of p-CuO and n-SnO_2_, gas molecules reacted with the oxygen species. Upon this reaction, the electrons were injected back to the conduction band of the oxide. Thus, the free electrons in SnO_2_ increased, which improved the oxide conductivity. On the other hand, for p-CuO, the injected electrons recombined with the holes, reducing the oxide conductivity. In addition, the transfer of electrons from p-CuO to n-SnO_2_ was not expected in the presence of ambient oxygen (in the air), and the transfer of electrons from SnO_2_ to p-CuO was not allowed because of the potential barrier. Thus, the effect of the heterojunction between p-CuO and n-SnO_2_ was negligible, and the sensor response here depended only on CuO and SnO_2_ bulks. Figure 6b shows the proposed band structure of Cu_x_Sn_1−x_O_2_, which shows an increase in the depletion layer with the existence of Cu ions. Due to the substitution of Cu in the matrix of SnO_2_, only one phase was formed. The substitution of Cu may have increased the active sites, which allowed the adsorbed oxygen molecules to capture more electrons from the conduction band of the oxide. This process increased the depletion layer of the oxide in the air. Upon exposure to the target gas, the oxygen species reacted with the gas to form H_2_O and CO_2_ and injected the electrons back to the conduction band [23,33,34], reducing the depletion layer. 

### 3.4. Effect of Concentration Ratio on Sensor Response 

The sensing properties of Cu_x_Sn_1−x_O_2_ and CuO/SnO_2_ sensors were investigated. The sensor signals toward CH_4_ gas for the sensors prepared with *x* = 0.00%, 0.25%, 0.50%, 1.00%, 2.00%, and 5.00% are shown in Figure 7a,b for operating temperatures of 250 and 300 °C, respectively. The sensor’s resistance decreased upon the exposure to CH_4_, suggesting that Cu_x_Sn_1−x_O_2_ is an n-type composite. Conversely, a p-type behavior was revealed for the sensor prepared by 10.00%, as shown in Figure 7c. This might be due to the dominance of the p-CuO phase on the sensing properties at low temperatures of 250 and 300 °C. However, n-SnO_2_ dominated the sensing properties at the higher temperatures of 350 and 400 °C, as shown in Figure 8a,b. The dominance of CuO or SnO_2_ seemed to be dependent on the operating temperature. This can be attributed to the surface activity of CuO at low temperature [21], while the SnO_2_ surface is much active at higher temperatures [4,5,6,8,9,10,11,12,15,16]. Thus, the sensor response decreased upon this sensing behavior, where the sensor response was positive for SnO_2_ and negative for CuO. The result also confirmed the interpretation mentioned in the above section.

The dependence of the sensor response on Cu concentration is given in Figure 9a. The sensor exhibited an improvement at low Cu concentrations. It firstly increased to the maximum response at *x* = 2.00%, then decreased gradually with a further increase of *x*. The result suggested that the optimal concentration of Cu embedded into SnO_2_ was about 2.00%. However, exceeding Cu concentration caused less sensitivity of SnO_2_. As mentioned above, the formation of CuO phase expressed a negative response behavior and reduced the sensitivity. Also, the sensing property was controlled by the bulk of CuO and SnO_2_ due to the large size of crystallites, resulting in insignificant heterojunction between these two phases. The response of the SnO_2_ sensor was obviously small compared to the Cu_x_Sn_1−x_O_2_ sensor. Therefore, the enhancement of sensor response toward CH_4_ gas was attributed to the Cu substitutions or the absence of bridging oxygen in the SnO_2_ matrix. Figure 9b shows the sensor response as a function of operating temperature. The sensor response firstly increased with the temperature, with the maximum response of 69.0 at 350 °C, and then decreased at higher temperatures. This indicated that the operating temperature of 350 °C was the optimal or near-optimal temperature of the presented sensors. The calculated average sensor response of all sensors as a function of temperature is shown by a dotted line in Figure 9b, with the sensing peak observed at 350 °C or close to it.

### 3.5. The Correlation of Sensing Properties and Surface Area 

The large surface area of nanomaterials is beneficial for sensing properties. Increasing the surface area may increase the active sites on the surface of nanomaterials. In the Brunauer–Emmett–Teller (BET) measurement, the adsorption isotherm curves (shown in Figure 10) were measured with N_2_ gas using a Quantachrome AUTOSORB-1-MP after the samples were dried in a vacuum at 150 °C for 1 h. The effective surface area of the composites synthesized of 0.25–10.00% Cu is shown in Figure 10b. The surface area first increased with the incorporation of Cu in the SnO_2_ lattice, reaching its maximal value of ~116 m^2^/g at 2.00% Cu content, indicating the development of more pores. Further increase of Cu decreased the surface area, i.e., ~33 m^2^/g for 10.00% Cu. Here, separated CuO and SnO_2_ nanorods were formed, and the crystallite size increased. Figure 10b demonstrates the correlation between the effective surface area and the sensing properties at different Cu concentrations. The most sensitive sensor fabricated of 2.00% Cu had the highest surface area. This large surface area may have allowed the diffusion of more gases to access more active sites. Sensors fabricated of nanocomposites with low surface area were less sensitive to the gas.

### 3.6. Calibration Curve and Sensor Stability

The response signal of the most sensitive sensor was measured four months later at various concentrations of CH_4_ (0.25–1.0%) and the operating temperature of 350 and 400 °C, and the results are shown in Figure 11a. The sensor response depended on the gas concentration, and it returned to the initial value upon switching off of the respective gas. From Figure 11a, one can observe that the sensor responded quickly to the change in the gas concentration. The sensor responded to the low concentrations of the gas with good sensitivity. Because this signal was measured four months after the first measurement, it can give us a good indication about sensor stability. Although it was measured after a long period, the sensor gave the same response value toward 1.0% methane; no draft was observed upon the exposure to various gas concentrations, and the resistance returned to its initial value after removal of the gas.

### 3.7. Response and Recovery Time Constants 

The calibration curve of the sensor fabricated of 2.00% Cu measured at 350 and 400 °C is presented in Figure 11b. The response shows a straight-line behavior for gas concentration of 0.00% up to 0.675%. The sensor response is represented by the empirical equation presented in the figure. A dramatic increase in the sensor response was observed for the higher gas concentration. The response is represented by a straight line with a large slope. There were two stages in the calibration curve of the present sensor, showing the nonlinear behavior in general. The increase in sensor response at 0.7% and 1.0% has also been detected in previous works for Co_3_O_4_ and In_2_O_3_ nanostructures [22,23]. This may be ascribed to an increase in CH_4_ pressure, allowing CH_4_ to diffuse deeply into the sensing layer and react with more oxygen species. However, the behavior is not well understood. Moreover, the gas concentration limit here was 1.0% in order to keep it less than the explosive limit of 5%. To better understand this behavior, a wide range of higher gas concentrations must be studied. We may carry out these measurements in the future with various materials supported with theoretical bases. 

The response and recovery time constants are essential parameters to estimate the sensing performance of the sensing layer on time. The rate of sensor response is described as the response or recovery time constants, which characterize the time required for the resistance to reach 90% of the equilibrium value after the gas injection and the time necessary for a sensor to attain a resistance 10% lower than its original value in the air, respectively. Figure 12a demonstrates the definition of the response and recovery time constants. The sensor responded quickly once the gas was introduced to the chamber. The time constants depended on the gas diffusion into the sensing layer, the surface temperature, and the speed at which the gas concentration filled the chamber. The time constants of the most sensitive sensor as a function of the operating temperature are shown in Figure 12b. This sensor had a shorter response–recovery time compared to the other sensors measured here. It is worth noting that the recovery time was much shorter than the response time at different temperatures. It also became shorter with increasing operating temperature, decreasing from ~85 s at 250 °C to 35 s at 400 °C. The response time did not depend on the operating temperature, keeping at ~149 s.

## 4. Conclusions

In summary, a novel Cu_x_Sn_1−x_O_2_ nanocomposite was synthesized by incorporating Cu^2+^ in the SnO_2_ matrix using cost-effective precursors and method. Different Cu concentrations were used to develop this composite to monitor the change in its structural and sensing properties. It was observed that, at low concentrations of copper precursor, only SnO_2_ phase was detected by XRD. The distribution of Cu in the SnO_2_ matrix was further measured by elemental analysis of EDX mapping. At high copper concentration, a separated monoclinic phase of CuO was formed. The formation of Cu_x_Sn_1−x_O_2_ or CuO phase was detected through the photoluminescent spectra of the radiative recombination. The effect of Cu concentration on the sensing properties of SnO_2_ toward methane (CH_4_) gas was also investigated. It was found that the sensor response was enhanced with increasing Cu^2+^ substitution in the SnO_2_ matrix, and the sensor embedded with 2.00% Cu exhibited an excellent sensitivity of 69.0 at 350 °C and short response–recovery time compared to other sensors reported here. However, the formation of separated CuO phase at 10.00% Cu negatively affected the sensing properties of the composite due to the competition between bulk materials of p-CuO and n-SnO_2_, which made the heterojunction at the interface between CuO and SnO_2_ insignificant. The present paper therefore provides an explanation for the improvement of sensing properties in terms of Cu substitution and the surface area.

## Figures and Tables

**Figure 1 sensors-19-02257-f001:**
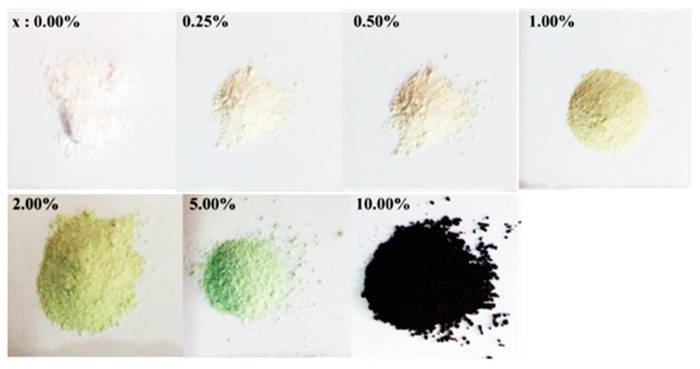
The change in product color with copper precursor *x*: CuCl_2_·2H_2_O.

**Figure 2 sensors-19-02257-f002:**
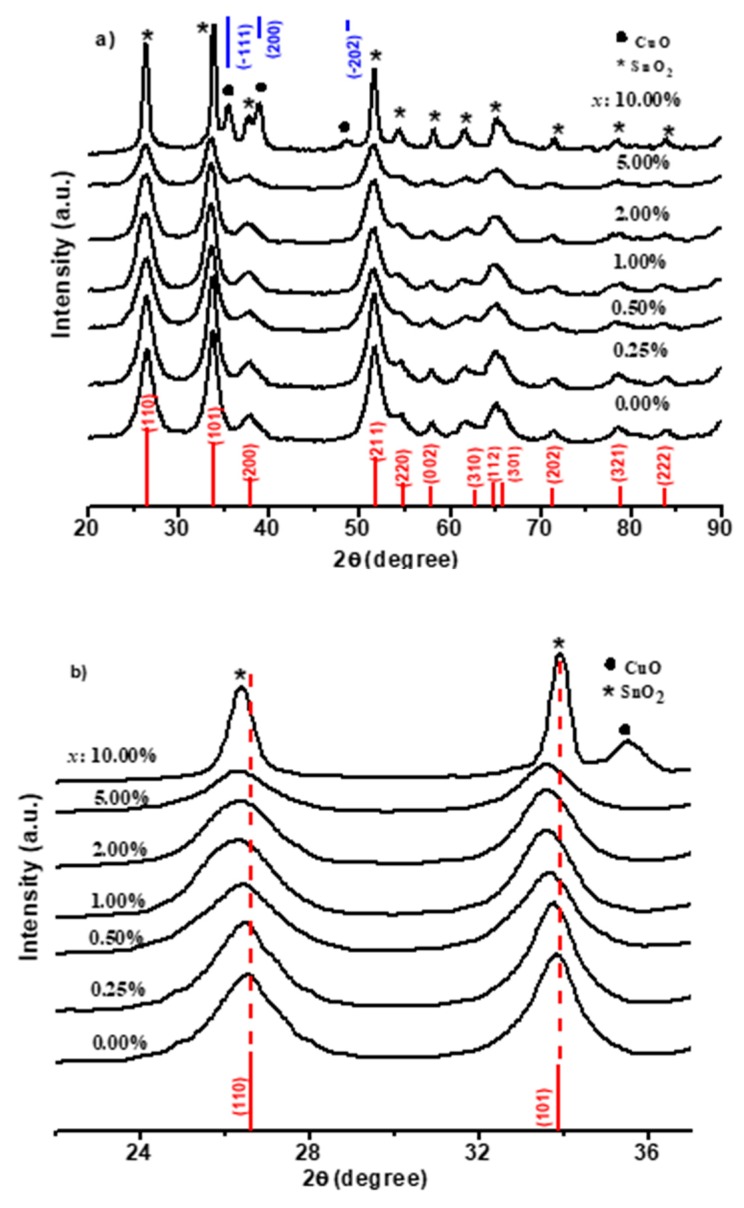
X-ray diffraction (XRD) patterns of the prepared composites: (**a**) 2Ѳ = 20°–90° and expanded figure (**b**) 2Ѳ = 20°–40°.

**Figure 3 sensors-19-02257-f003:**
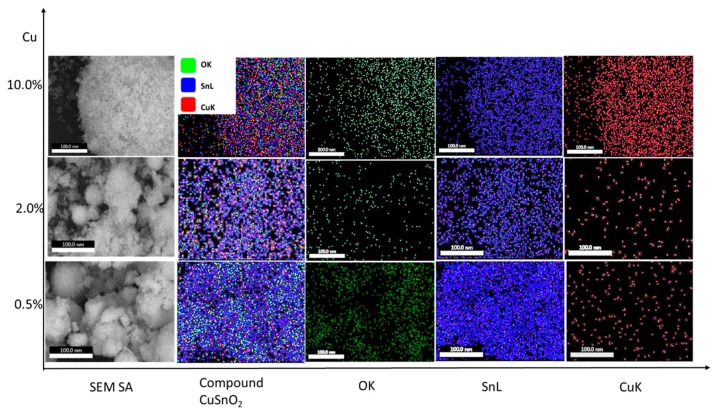
EDX element mapping of a selected area of Cu_x_Sn_1−x_O_2_ for SnL and CuK.

**Figure 4 sensors-19-02257-f004:**
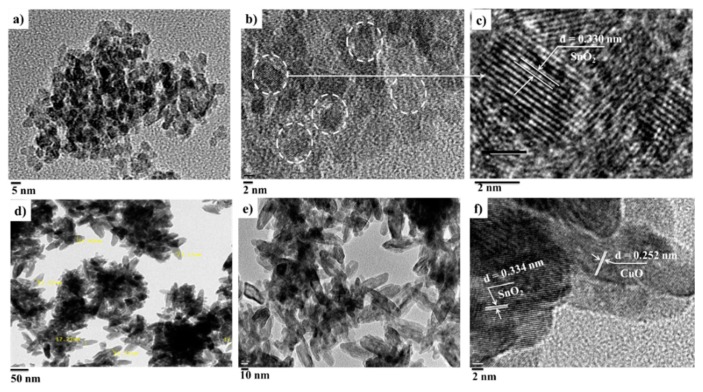
Typical high-resolution transmission electron microscopy (HRTEM) images of (**a**–**c**) *x* = 2.00% (Cu_x_Sn_1−x_O_2_) and (**d**–**f**) *x* = 10.00% (CuO/SnO_2_).

**Figure 5 sensors-19-02257-f005:**
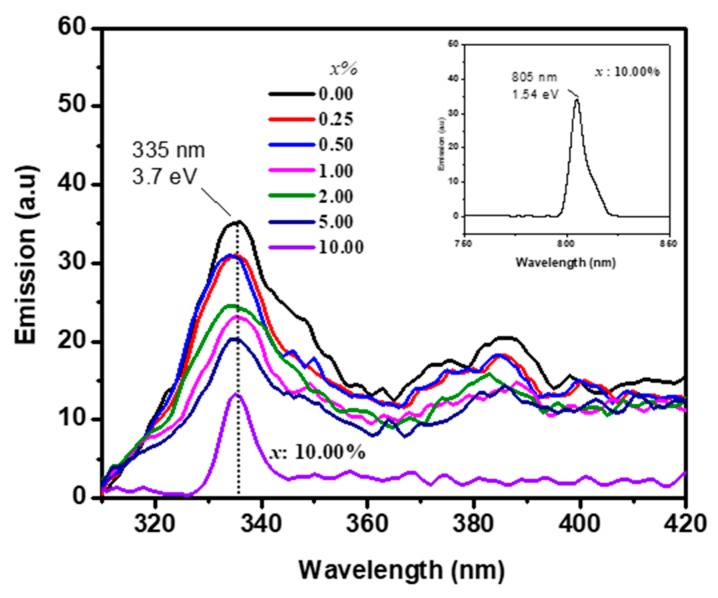
Fluorescence (FL) emission spectra at an excitation wavelength of 290 nm and 600 nm (inset) for *x* = 0.25–10.00%.

**Figure 6 sensors-19-02257-f006:**
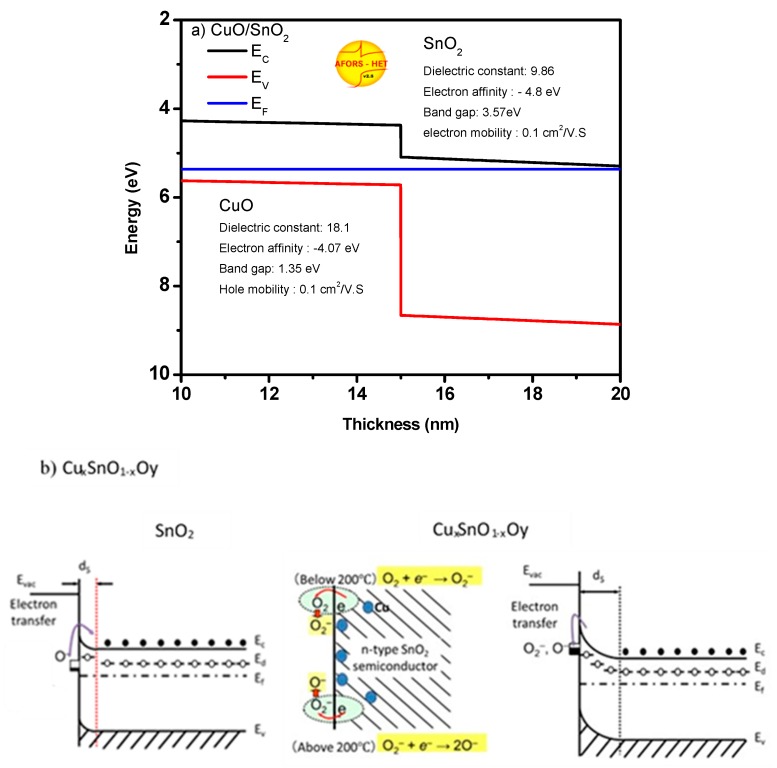
The possible gas-sensing mechanism of Cu_x_Sn_1−x_O_2_ and CuO/SnO_2_ nanostructures: (**a**) heterojunction of CuO/SnO_2_ and (**b**) band structure of Cu_x_Sn_1−x_O_2_. Note: the depletion layer increased by the substitution mechanism has been taken into consideration.

**Figure 7 sensors-19-02257-f007:**
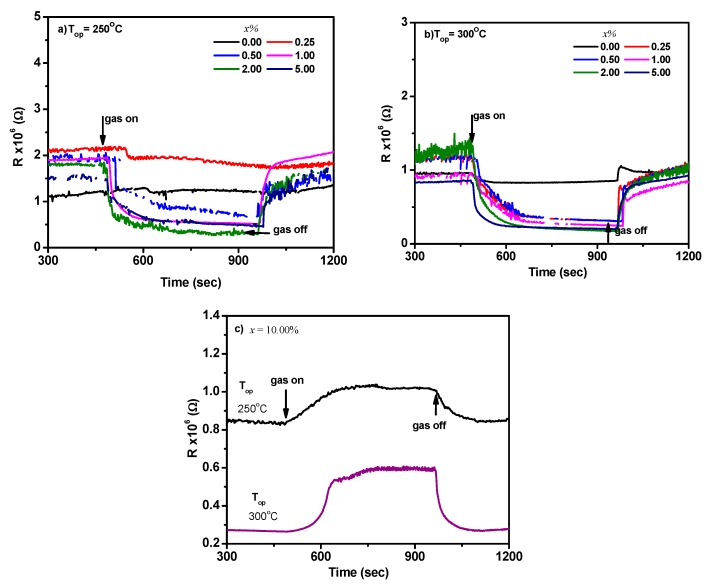
Sensor signal upon exposure to 1.0% CH_4_ gas for (**a**) *x* = 0.00–5.00% at 250 °C, (**b**) *x* = 0.00–5.00% at 300 °C, and (**c**) *x =* 10.00%. Note: the change in resistance of the sensor fabricated of 10.00% showed p-type semiconductor at these operating temperatures.

**Figure 8 sensors-19-02257-f008:**
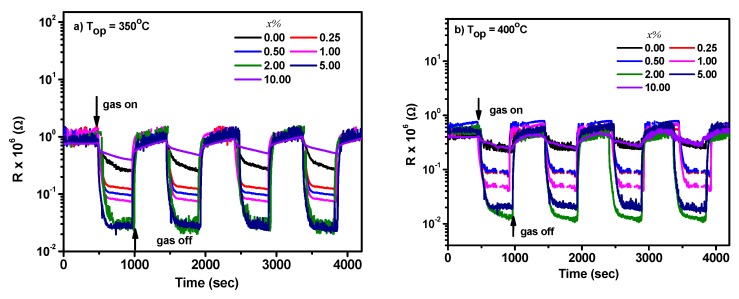
Sensor response toward 1.0% CH_4_ for the prepared composites at *x* = 0.00–10.00% at (**a**) 350 °C and (**b**) 400 °C. Note: the sensor fabricated of 10.00% showed an n-type semiconductor.

**Figure 9 sensors-19-02257-f009:**
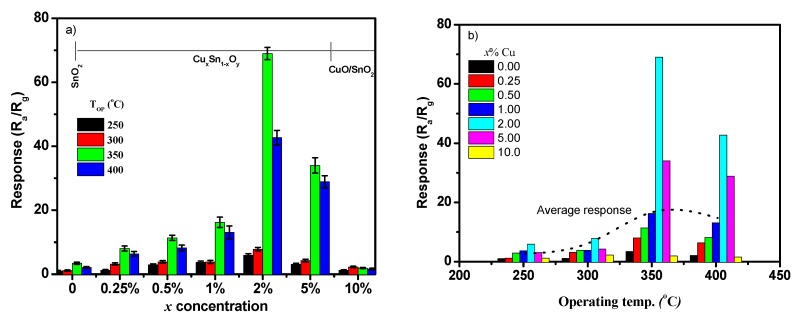
(**a**) Sensor response versus Cu concentration of 0.00–10.00%; (**b**) sensor response versus operating temperatures of 250–400 °C. Note: the dotted line is only a guide for the average sensor response for all sensors.

**Figure 10 sensors-19-02257-f010:**
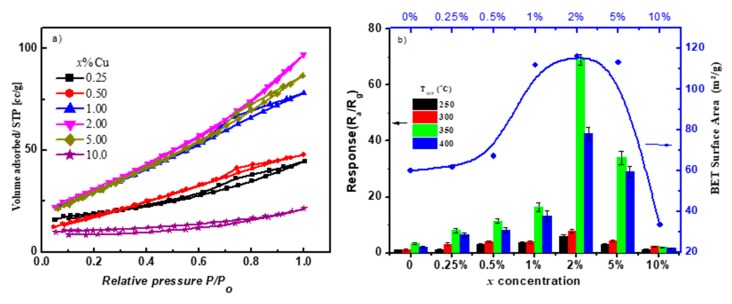
(**a**) Adsorption–desorption curves for Brunauer–Emmett–Teller (BET) measurements; (**b**) BET calculated surface area versus the Cu concentration, attached with the sensor response.

**Figure 11 sensors-19-02257-f011:**
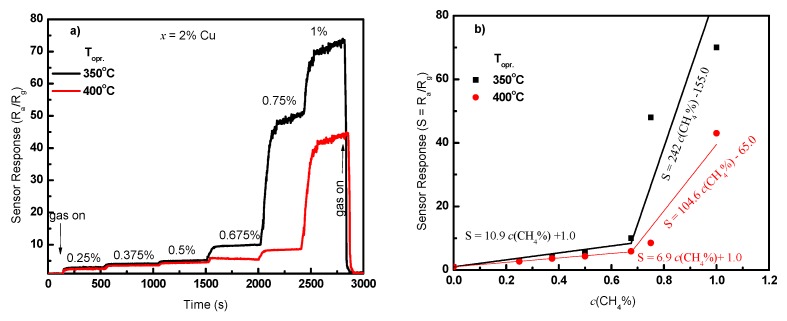
(**a**) Sensor response for various gas concentrations and (**b**) calibration curve for the sensor fabricated of 2.00% Cu at two different operating temperatures.

**Figure 12 sensors-19-02257-f012:**
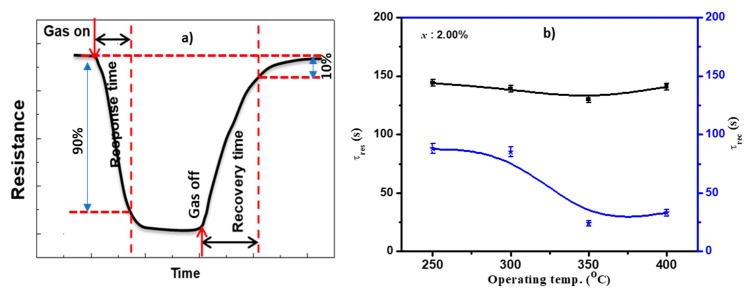
(**a**) Diagram illustrating the definition of the response and recovery time constants and (**b**) the calculated response and recovery time constants of 2.00% Cu sensor as a function of operating temperature.

**Table 1 sensors-19-02257-t001:** CH_4_ sensors based on undoped and doped SnO_2_ materials reported in the literature and in the present study.

Sensing Material	Operating Temperature (°C)	CH_4_ Concentration %	Response	Ref.
SnO_2_	350	0.200	1.35	[4]
Ag–SnO_2_	430	0.200	1.75	[5]
Fe–SnO_2_	350	0.025	1.30	[6]
porous 3D SnO_2_	120	0.050	1.80	[7]
Pt–SnO_2_	350	0.100	4.50	[8]
Pt–SnO_2_	400	0.100	1.55	[9]
Pd–SnO_2_	400	0.660	20.00	[10]
Pd–SnO_2_	350	0.100	1.35	[11]
Pt/SnO_2_	400	0.500	30.00	[12]
SnO_2_ NR–NP–Gr hybrids	150	1.000	50.00	[13]
PdPt–SnO_2_–rGO	150	0.100	69.50	[14]
Sn_0.9-x_In_0.1_Cu_x_O_2_	400	0.250	9.00	[15]
SnO_2_–In_2_O_3_–TiO_2_/MgO	300	0.500	2.40	[16]
CuO NCs	200	1.000	2.22	[21]
Co_3_O_4_ NPs	200	1.000	1.28	[22]
Cu_0.02_Sn_0.98_O_2_ nanocomposite	350	0.250	3.50	Present
		0.375	4.50	
		0.500	5.50	
		0.675	10.00	
		0.750	50.50	
		1.000	69.60	

**Table 2 sensors-19-02257-t002:** The value of the molar fractions of Cu and Sn in Cu_x_Sn_1−x_O_2_ and CuO/SnO_2_ nanostructures using X-ray fluorescence (XRF) and elemental analysis of OK, SnL, and CuK using energy-dispersive X-ray (EDX) technique.

Intended Molar Fraction (*x* %)	X-ray Florescence Molar Fraction%		EDX Measurements							
	Cu%	Sn%	OK		SnL		CuK		Cu/Sn	O/(Cu + Sn) at.
			wt.%	at.%	wt.%	at.%	wt.%	at.%	at.	
0.00	0.000	100.000								
0.25	0.188	99.811								
0.50	0.472	99.527	16.320	58.1600	79.760	38.320	3.920	3.520	0.092	1.386
1.00	0.941	99.058								
2.00	2.342	97.657	7.950	36.510	79.900	49.450	12.140	14.040	0.283	0.575
5.00	4.500	95.499								
10.00	11.087	88.912	20.440	54.870	52.050	35.180	27.510	9.950	0.282	1.219

**Table 3 sensors-19-02257-t003:** Average crystallite size and lattice parameters of Cu_x_Sn_1−x_O_2_ and CuO/SnO_2_ nanostructures.

XRD Analysis			
Sample *x*%	Average Crystallite Size (nm)	Lattice Parameter *a*, c (Å)	Unit Cell Volume (Å^3^)
0.00	5.90	4.7406, 3.1676	72.19
0.25	5.82	4.7533, 3.1956	72.20
0. 50	4.80	4.7529, 3.1965	72.21
1.00	4.74	4.7566, 3.1919	72.22
2.00	4.70	4.7566, 3.1927	72.24
5.00	4.69	4.7643, 3.1894	72.39
10.00	15.28	4.7616, 3.1896	72.32

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
