# Peer review of "Preparation and Characterization of Developed CuxSn1−xO2 Nanocomposite and Its Promising Methane Gas Sensing Properties"

_sensors, 2019, doi:10.3390/s19102257_

Round 1

Reviewer 1 Report

In this work authors described with great detail the process to synthesize a Cu_xSn_{1-x}O_2 structure based on Tin (II) chloride [1-x: SnCl4.5H2O] and Copper chloride dehydrate [x: CuCl2.2H2O] as a dopant source were dissolved in distilled water of 10.0 ml volume. Here authors varied the concentration of x looking for the optimum value that can be more sensitive to CH4, and they found that it is around 2%.

However, authors did not explain with detail the sensing characteristics of their structures. From my point of view authors focus the discussion of the manuscript on the synthesis process of CuxSn1-xO2 and just give a very superficial description of the sensing characteristics. I consider that this information it is quite important in a sensors’ related paper. Moreover, I strongly recommend to authors to sort the information, for instance in one of two sections discuss the synthesis process and later in another one of two sections describe your sensing setup, the sensing performance of your structures and the advantages/disadvantages of use your structures. For instance, your current section 3.4 is allocated after the sensor results are presented, but in this section authors described the sensing mechanism of the composites. So, I consider that for instance, this can be placed before the sensors sections, as an introduction to you sensing “devices”. Separating the information can be of great help to readers.

 Moreover, right now it is quite difficult to visualize what are the advantages of your structures in gas sensing applications, are these more sensitive?, are robust to water vapor or moisture? The time response is faster?, the CH4 concentration dynamic range is broad enough?, what about the sensitivity is higher?. All these characteristics, must be compared with other CH4 sensors in order to give to the reader an idea about the potential that have these composites in sensing applications. Therefore I suggest to reconsider the manuscript after a major revision is carried out. These corrections must attend the points previously expressed and additionally must address the following:

1.       Authors mentioned the response and recovery time constants, but never defined them neither explained how you measured to build the figure 11.

2.       Define BET

3.       In figure 9, authors talk about Adsorption-desorption curves for BET measurements, however in the next paragraph fails to relate this data with the sensor response. Authors must relate, clearly and briefly. As is a methane sensor I would like to know the behavior of the sensor to this gas, for instance, how efficient is in the sensor in the absorption and desorption processes. Is there a concentration level for which the sensor get “poisoned”?.

4.       Authors showed the sensor response up to 1% CH4 in figure 10. However it has a nonlinear behavior, and it get stronger for 0.7% CH4. Please discuss how this issue can affect the rest of the sensor parameters, such as the sensitivity, the resolution, the CH4 concentration range of the sensor (it must be ranges for which the sensor has a better sensitivity), d) the stability of the sensor and how this affect or limit the sensitivity, etc.

5.       I recommend that for your sensing sections look for further references, these can be helpful to compare your sensing system with other sensors based on different materials and even based on different technologies. In order to have a broader scope and authors can stress the advantages. In the MDPI journals (ie. Sensors, Applied Sciences, Chemosensors) there are recent open source papers, about gas sensing, not necessarily must be CH4 sensors, but the methodologies and figures of merit and how these can be calculated can be of relevance to complete the analysis.

6.       In line 89, please ckeck if Jasko is correct or it must be Jasco.

Author Response

Dear Prof. 

We thank the reviewer for cutting part of his to revise our manuscript and give us valuable comments. The manuscript has been reconsidered according to to the reviewer comments. 

Reviewer 2 Report

References 21 and 22 are irrelevant to what is postulated in the introduction. ref. 21 describes DECREASE of oxygen vacancy conc. during modification of brookite (a very different compoud compared to cassiterite), ref.22 does not have any considerations on the correlations between copper modification and SnO2 oxygen vacancy concentration.

The authors need to find another explanation to the observed effect of sensor response improvement towards methane as the idea on oxygen vacancy concentration increase is not supported by any of the used methods. The EDX technique is not suitable for that - it does not work good towards light elements and usually considered as semi-quantitative. Moreover it does not allow to make any judgements on crystallographic positions of the elemens under investigations.

The incorporation of low valence substitutional defect (such as Cu (II))  into the structure of SnO2 in Sn (IV) position is unlikely to bring any new oxygen vacancies. The opposite effect is expected. Authors should provide a quasi-chemical equation in Kröger–Vink notation for this defect formation process and it will become clear.

The data on BET surface area may be a good explanation on the response growth for obtained nanocomposites.

The use of English should be seriously improved as well.

Author Response

Dear Prof. 

We thank the reviewer for cutting part of his time to revise our manuscript and give us valuable comments. The manuscript has been reconsidered according to the reviewer comments. 

Reviewer 3 Report

The author has made better modifications, but there are two problems that the author has not solved well.

1. Selectivity and stability are very important parameters for evaluating the quality of a sensor. I hope that the author can objectively evaluate the performance of the sensor you have prepared.

2. I asked why the formation of CuO/SnO2 heterojunction would increase the resistance. I hope the author can explain it through the resistance variation diagram or by citing similar literature. Don't avoid the problem.

Author Response

Dear Prof. 

We thank the reviewer for cutting part of his time to revise our manuscript and give us valuable comments. The manuscript has been reconsidered according to to the reviewer comments. 

Round 2

Reviewer 1 Report

Authors have enhanced the manuscript by taking into consideration most of the comments of the reviewers. Therefore I would like to suggest to accept the paper after minor revisions.

1.       I would like to suggest to incorporate in the manuscript as Figure 12a the figure presented in the point 1 of the cover letter. This for clarity purposes. Your current Figure 12 can be the 12b.

2.       Please give a detailed review to the grammar/typos to clean some English edition errors.

Author Response

Dear Dr. Julia Zhu, 

Dear Dr. Charlene Dong        

Editor of Sensors,

Good day,        

Thank you very much for sending me the comments of the reviewer on our manuscript. We have carefully reviewed our manuscript mentioned above, and please accept my deepest thank to the Reviewers for the valuable comments and suggestions which have resulted in a large improvement of the manuscript. We really grateful and appreciate the efforts for the reviewers and journal staff. 

Please find enclosed the attached of the re-revised manuscript, enclose to response to the Reviewer’s comments.

Reviewer 2 Report

eqs 1 and 2 are wrong. they are not electronically balanced 

conclusions on creation of oxygen vacancies are not supported by the experimental data

particularly, increase in oxygen concentration vacancies should improve the materials conductance, which is not observed. actually, the opposite effect can be seen on fig.6-8

the paper should be rewritten in this context.

Author Response

 Editor of Sensors,

Good day,        

Thank you very much for sending me the comments of the reviewer on our manuscript. We have carefully reviewed our manuscript mentioned above, and please accept my deepest thank to the Reviewers for the valuable comments and suggestions which have resulted in a large improvement of the manuscript. We really grateful and appreciate the efforts for the reviewers and journal staff. 

Please find enclosed the attached of the re-revised manuscript, enclose to response to the Reviewer’s comments.

Round 3

Reviewer 2 Report

now the paper is consistent

This manuscript is a resubmission of an earlier submission. The following is a list of the peer review reports and author responses from that submission.

Round 1

Reviewer 1 Report

Authors describe the process to synthesize a CuxSn1-xO2 structure based on Tin (II) chloride [1-x: SnCl4.5H2O] and Copper chloride dehydrate [x: CuCl2.2H2O] as a dopant source were dissolved in distilled water of 10.0 ml volume. Here authors varied the concentration of x in order to determine the effects of varying the Cu concentration over the CuxSn1-xO2 structure and its sensing characteristics. However, authors failed to show the sensing advantages of the sensor since the description of the experiments and the methodology followed to calculate the resistivity, time response, CH4 concentration dynamic range, etc., are quite poor and incomplete. Therefore I suggest to reject the manuscript. Some specific aspects that I consider must be described in the manuscript are the following.

1.       In Figure 5 it is shown the behavior of the resistance of different samples of CuxSn1-xO2 to the presence to methane. Here it is interesting to observe the changes of the different structures (due to different Cu concentrations), however authors did not specified the CH4 concentration used to carry out these measurements.

2.       Moreover, authors did not discuss about the stability and hysteresis of the structures, for instance in figures 5a and 5b, it can be observed that the resistivity is always changing even when there is not CH4, and also it can be observed that there are cases that after the gas flow is off the resistivity did not return to their initial value (in synthetic air). These issues are quite important for sensing applications and based on these figures it is not clear that these structures can provide good sensing characteristics.

3.       In figures 6a and 6b it was not mentioned what concentration of CH4 was used to carry out the experiment. Moreover it is not clear why did you used log scale for these figures when in Fig 5 it was used a linear scale. I consider that for ease the comparison all similar figures must be in the same scales.

4.       Finally authors jumped into the conclusion that the sensitivity of the CuxSn1-xO2 nanoparticle, when x: 2.00%, is as high as 69. Here I consider that to provide this kind of figures of merit it is necessary to perform the same experiments but with different CH4 concentrations levels, to determine for instance: a) if the sensor can detect the changes of CH4 concentrations, b) determine if the sensitivity is linear or non linear versus the CH4 concentration, c) in what CH4 concentration ranges the sensor is more sensitive, d) the stability of the sensor and how this affect or limit the sensitivity, etc. 

5. About the edition I suggest to authors to prepare in just one figure all the subplots in order to ease the reading of the manuscript (for examples Fig5 and Fig. 6).

Reviewer 2 Report

Copper modified nanocrystalline SnO2 has been reported previously (eg.  Krivetskiy V, Ponzoni A, Comini E, Badalyan S, Rumyantseva M, Gaskov A (2010) Selectivity Modification of SnO2-Based Materials for Gas Sensor Arrays. Electroanal 22 (23):2809-2816 and references therein), so the novelty of the research needs to be stated more clearly.

Notes:

1) In abstract (Line 17) in situ synthesis is mentioned, however the described technique has nothing to do with in situ methods

2) SnO2 sensors are widely used for flammable gases detection, what drawbacs (page 1 line 37) are mentioned is not clear and how Cu modification is supposed to fix them is not explained by the authors.

3)page 2 Line 57 tin has valence IV  in tin chloride,  not II as written

4)sensor fabrication and the means of their temperature control are not described

5)line 78 XRF technique is useless to proove the incorporation of Cu in SnO2. It can only say that copper is present in the nanocomposite structure not SnO2 lattice

6) the method of lattice parameter calcuclations is not given

7)page 5 - the fade of fluorescence during Cu content growth in SnO2 is not discussed, what causes it (Fig. 4)

8) fig 6 twice

9)authors say that Cu incorporation causes increases oxygen vacancy concentration. No quasi chemical equation is given for this process. Also this ptocess should give rise to the materials resistance drop. It is not observed from figs 5,6 and 7

10) the explanation of the electrical or catalytic effect of Cu is insufficient without data on the efective surface area of the synthesized samples. this data should be collected and presented.

11) line 188 page 10 - unit cell is said to shrink by the authors, but the table 1 indicates the counter effect. 

12) line 191 and 192 - electron donor efect of Cu is not supported by the resistance, requires other evidence as well as catalytic effect

Reviewer 3 Report

1. Language needs extensive revision.

2. Title does not match with the developed work. The manuscript does not discuss the analysis of actual applications and performance changes after the actual application.

3. The author should revise the abstract to make the content more concise and clear.

4. More analysis should be used to confirm the composition, for example what about EDX, EDS analysis? or example what about EDX, EDS analysis ? Is there any data on the STEM-mapping to confirm the material composition and atom distribution?

5. The introduction part is a lack of information and background about the other materials and approach that used in this field. Thus, it is better to provide more information about other materials that used in this field of applications, different application of the current material, and advantage of the current study.

6. What is the concentration of the added NaOH solution in the experimental part? Is SnCl4.5H2O and CuCl2.2H2O compounded by mass ratio or molar ratio?

7. What is the basis of crystal size calculation? Such as the use of which formula, method.

8. I did not find the diffraction peak of the (211) plane in Figure 2b.

9. Figure 5 does not correspond to the 200 °C resistance change diagram, but the article has carried out a corresponding analysis.

10. Figure 8 should be better explained, and the authors write that the response-recovery time is shorter than other documents, but we did not see the comparison, I think it is not convincing to say this directly.

11. The working temperature point of the study should be more dense, and perhaps the optimal operating temperature of the sensor is not as high as 350 °C.

12. As can be seen from Table 1, although the material you synthesized has a high response to methane compared with other literature reports, the detection temperature and concentration are also high, which is very disadvantageous from the practical point of view.

13. The response value is important, but the basic gas sensitivity of a sensor also includes: selectivity, stability, detection limits, and so on. I think the author should evaluate the sensor in this aspect.

14. The mechanism part should be modified to confirm that the explanation is correct, for example: a. the oxygen molecule captures free electrons from the oxide conduction band rather than the surface, b. 7he electron affinity should be represented by a negative sign, and so on. At the same time, please explain briefly why the formation of CuO/SnO2 heterojunction will increase the resistance.

15. Please make corresponding changes to the conclusions according to the previous revision.

16. Please correct references with orderly.